# Dental Health Services Response to COVID-19 in Norway

**DOI:** 10.3390/ijerph17165843

**Published:** 2020-08-12

**Authors:** Lina Stangvaltaite-Mouhat, Marte-Mari Uhlen, Rasa Skudutyte-Rysstad, Ewa Alicja Szyszko Hovden, Maziar Shabestari, Vibeke Elise Ansteinsson

**Affiliations:** 1Oral Health Centre of Expertise in Eastern Norway, Sørkedalsveien 10A, 0369 Oslo, Norway; linas@viken.no (L.S.-M.); martemariu@viken.no (M.-M.U.); rasas@viken.no (R.S.-R.); ewah@viken.no (E.A.S.H.); maziars@viken.no (M.S.); 2Department of Clinical Dentistry, Faculty of Health Sciences, UiT the Arctic University of Norway, N-9037 Tromsø, Norway

**Keywords:** COVID-19, SARS-CoV-2, dentistry, dental public health, urgent dental care, emergency dental treatment, risk perception, preparedness

## Abstract

We aimed to investigate the management of urgent dental care, the perception of risk and workplace preparedness among dental staff in Norway during the COVID-19 pandemic. An electronic questionnaire regarding the strictest confinement period in Norway (13 March–17 April 2020) was distributed to dental staff. Among the 1237 respondents, 727 (59%) treated patients, of whom 170 (14%) worked in clinics designated to treat patients suspected or confirmed to have COVID-19. Out of them 88% (143) received training and 64% (103) simulation in additional infection prevention procedures, while 27 (24%) respondents reported deviation. In total, 1051 (85%) respondents perceived that dental staff had a high risk of being infected, 1039 (84%) that their workplace handled the current situation well, 767 (62%) that their workplace had adequate infection control equipment and 507 (41%) agreed that their workplace is well equipped to handle an escalation. Before an appointment, 1182 (96%) respondents always/often inquired per phone information if a patient experienced symptoms of COVID-19, and 1104 (89%) asked about a history of travel to affected areas. Twice as many patients on average per week were treated by phone than in a clinic. A lower proportion of dental staff in high incidence counties applied additional infection prevention measures compared to low and medium incidence counties. To conclude, urgent dental health care was managed relatively well in Norway. Additional training of the dental staff in adequate infection prevention and step-by-step procedures may be needed. These results may be used to improve the dental health service’s response to future outbreaks.

## 1. Introduction

Coronavirus disease 2019 (COVID-19) is a public health emergency of international concern announced by the World Health Organization on 30 January 2020 and declared a pandemic on 11 March 2020 [1,2]. COVID-19 is caused by a novel coronavirus named “SARS-CoV-2”, which belongs to severe acute respiratory syndrome coronaviruses (SARS-CoVs) [3]. This is the third outbreak of an infection caused by a coronavirus in less than 20 years. The severe acute respiratory syndrome (SARS) outbreak in 2002–2003 resulted in more than 8000 cases in 26 countries, and had a mortality rate of approximately 10% [4,5,6,7]. In 2012–2013 the outbreak of Middle East respiratory syndrome (MERS) spread, and up to date 27 countries reported 2500 confirmed cases with a 34% mortality rate [8,9,10,11]. 

To date (10 August 2020), there have been 9468 confirmed cases of COVID-19 and 256 deaths in Norway [12]. SARS-CoV-2 can be transmitted by two main routes: respiratory and contact. Respiratory droplets are generated when an infected person coughs or sneezes. Transmission by direct contact occurs through skin contact followed by touching the oral, nasal or ocular mucous membranes. The virus may also be transmitted by indirect contact via objects and surfaces [13]. Recent evidence suggests that SARS-CoV-2 is detected in saliva [14], can be transmitted by aerosol- generated procedures [15] and from asymptomatic patients [16].

In Norway, during the period of containment for COVID-19 (13 March–17 April 2020) dental health services suspended routine non-urgent dental health care. Public dental health services in Norway correspond to around 30% of the total dental health service, and twice as many dentists work in private dental clinics. It is unknown how many private clinics complied with the recommendation to suspend non-urgent dental care. Health care professionals in both public and private practice are required by the Norwegian law to provide emergency health care to all patients. Indeed, patients confirmed or suspected to have COVID-19 have the same right to emergency care as non-infected patients. However, to provide dental care during the pandemic required an extra focus on protective measures and personal protective equipment (PPE). In the period between 13 March–17 April 2020 there was a shortage of PPE in the health service in Norway; consequently, some dental clinics had to be closed or staffed down in this period. On the 14 March 2020, the Directorate of Health in Norwayrequested the dental public sector to establish an emergency service for patients suspected or confirmed to have COVID-19. This could be in collaboration with the private sector and/or the universities, however, to our knowledge, the majority of the counties selected a fewpublic clinics in each region that was prepared and designated for this purpose subsequently. 

Dental staff may be at high risk of being infected by COVID-19, as the practice of dentistry involves the use of rotary and surgical instruments (e.g., handpieces or ultrasonic scalers) and air-water syringes. These instruments create a visible spray that contains droplets of water, saliva, blood, microorganisms and other debris. Aerosols may also be generated [17]. Dental staff may become potential carriers of the virus and, if adequate precautions are not taken, the dental office can potentially serve as a cross-infection location [18]. Since dental settings have unique characteristics, they warrant specific additional infection control considerations. 

Before the present study was launched, the US Centers for Disease Control and Prevention (CDC) had released Interim Infection Prevention and Control Guidance for Dental Settings during the COVID-19 Response [17] and the World Health Organization (WHO) had released its Guidance for Health Workers during Coronavirus Disease (COVID-19) Outbreak, without specifying dental settings [19]. Due to lack of international guidelines, national recommendations had to be devised for additional infection control and urgent dental care. Since Norway has not been affected by previous coronavirus outbreaks (SARS or MERS), this emergency was unprecedented for Norwegian dental staff. Therefore, the aim of this case study was to investigate how urgent dental health care was managed in Norway, what additional infection prevention and control measures were employed by dental staff and to assess the dental staff perception of risk and workplace preparedness. 

## 2. Materials and Methods

### 2.1. Study Design and Participants

The present study comprised a cross-sectional questionnaire survey among dental staff in Norway. The questionnaire was sent electronically via QuestBack to chief dental officers in counties who were asked to distribute the questionnaire among dental clinics in Norway. All dental staff, including specialists, general dental practitioners, dental hygienists and dental assistants were invited to fill in the questionnaire. Invitations to dentists in the private sector were distributed via local associations of the Norwegian Dental Association (NDA). The questionnaire was sent out 4 May 2020 to the public sector and 15 of May 2020 to the private sector and asked information related to the strictest confinement period in Norway(13 March–17 April 2020). Reminder for participation was sent three times and the questionnaire was closed on 26 June 2020.

### 2.2. Questionnaire

The self-reported questionnaire was based on information provided by CDC, WHO, Norwegian Institute of Public Health, Ministry of Health in Norway, and guidelines provided by the Norwegian counties. The questionnaire consisted of four parts: (I) Background characteristics, (II) Dental health service management, including treatment of patients suspected or confirmed to have COVID-19, (III) Dental staff perception of risk and preparedness, and (IV) Psychological impact. The present article intended to report the results from the first three (3) parts of the questionnaire. The background characteristics included information about sex, age, work experience in years, profession (specialist/general dental practitioner, dental hygienist, dental assistant), area of the dental clinic (urban, which had >50,000 inhabitants, peri-urban 5000–50,000 inhabitants, rural <5000 inhabitants), size of dental clinic (large, ≥7 employees and small <7 employees), sector of main workplace (public, private), and if the respondent worked clinically with patients during COVID-19 outbreak (yes/no). Dental health service management part asked information about triage of patients per phone, additional infection control measures, three (3) most common treated conditions, if a clinic was eligible to treat patients suspected or confirmed to have COVID-19, and knowledge of where to refer patients with urgent needs who are suspected or confirmed to have COVID-19. Questions related to treatment of patients confirmed or suspected to have COVID-19 inquired information about the number of patients treated, additional infection control measures and procedures, if dental staff were trained to follow them and if there was a deviation, if scientific information was available and from where. Regarding perceptions, dental staff was asked on a 5-point Likert scale (1_completely agree, 5_completely disagree) to assess four statements: dental staff risk to be infected; if workplace had adequate infection control equipment; how workplace handled the current situation; if workplace was well equipped to handle an escalation. For statistical analyses the responses were dichotomized into agree (points 1 and 2) and disagree (points 3–5).

The questionnaire was face validated by several experts in the field and pre-tested by 10 dentists, which were not included in the analysis. 

The incidence of cases in counties was retrieved from Norwegian Institute of Public Health, and subsequently the counties were grouped into low incidence counties (<100 reported cases per 100,000), medium incidence counties (100–150 reported cases per 100,000) and high incidence counties (>150 reported cases per 100,000) for statistical analyses [20]. 

### 2.3. Statistical Analysis

Statistical Package for the Social Sciences (SPSS) version 26.0 (IBM SPSS, Armonk, NY, USA) was used for statistical analyses. The chi-square test and analysis of variance (ANOVA) with Tukey adjustment were used to identify differences in characteristics between strata. Univariable and multivariable binary logistic regression analyses were used to assess the association between the perception of risk and workplace preparedness (four (4) outcomes) and potential determinants. Variables significantly associated with the outcome in bivariate analyses at *p*-value < 0.2 were entered into the regression analyses as independent variables. The results were presented as odds ratios with 95% confidence intervals (CI). The statistical significance was set at *p* < 0.05.

### 2.4. Ethical Considerations

Approval was obtained from the Norwegian Centre for Research Data (907304). Voluntary participation was based on a signed written informed consent.

## 3. Results

There was an overrepresentation by females, 1106 (89%), and those working in public service, 1134 (92%). Out of all the respondents, 590 (48%) were dental specialists/general dental practitioners, 235 (19%) were dental hygienists and 412 (33%) were dental assistants. Seven hundred and twenty-seven (59%) respondents worked with patients during the strictest confinement period 13 March–17 April 2020, in Norway (413 (70%) dental specialists/general dental practitioners, 66 (28%) dental hygienists and 248 (60%) dental assistants) (Table 1). 

Table 2 shows the results regarding organization of urgent dental care in oral health service and management of patients NOT suspected to have COVID-19. The majority of the dental staff always/often inquired information per phone if a patient experienced symptoms of COVID-19 or had a history of travel to affected areas (1182 (96%) and 1104 (89%), respectively). A significant difference was observed among county incidence categories. Dental specialists/general dental practitioners on average per week treated five (standard deviation (SD) 4.6) patients not suspected to have COVID-19. On average per week 11 (SD 13.0) patients were clarified per phone out of whom three (SD 4.3) received drug treatment. Dental specialists/general dental practitioners were asked to rank three most common conditions the patients had during the period 13 March–17 April 2020. Out of 440 (35%) clinicians who responded, the most common urgent conditions were severe dental pain from pulpal inflammation (321, 73%), abscess or localized bacterial infection resulting in localized pain and swelling (264, 60%) and pericoronitis or third-molar pain (233, 53%) (data not shown). 

When treating patients not suspected to have COVID-19, 389 (88%) of dental specialists/general dental practitioners) reported to follow additional infection prevention and control measures. The most common disinfection product was 70% ethyl alcohol; there was a significant difference in the products for disinfection between counties The majority of respondents used mouth rinse and high-volume suction as an additional protective measure while treating patients, while less than half used rubber dam; a significant difference in use of these additional protective measure was observed among county incidence categories (see Table 2). 

Out of the respondents who were not from clinics designated to treat patients suspected or confirmed to have COVID-19 (1067, 86%), 1064 (99%) were aware where to refer a patient suspected or confirmed to have COVID-19 for emergency treatment, or where to find such an information; there was a significant difference among county incidence categories (see Table 2).

Table 3 shows the results of the organization of urgent dental care for patients suspected or confirmed to have COVID-19. Out of all the respondents, 170 (14%) were from clinics designated to treat patients suspected or confirmed to have COVID-19; out of them 72 (42%) were dental specialists/general dental practitioners, 28 (17%) dental hygienists and 70 (41%) dental assistants. Very few patients suspected or confirmed to have COVID-19 were treated in the designated clinics. The majority of the dental staff (67, 39%) reported to leave the room between 35 min and 3 h in between such patients; there was a significant difference among county incidence categories. Out of the dental staff working in clinics designated to treat patients suspected or confirmed to have COVID-19, up to 20% reported not to have available respirators FFP2 or FFP3 standard or equivalent, gowns and aprons in their workplace; there was a significant difference among the county incidence categories. The majority of dental staff received training in additional infection prevention and control procedures either digitally or in a clinic, and mostly guidelines developed by county (84, 49%) and university (59, 35%) were followed. The majority of dental staff reported that their clinic developed step-by-step procedures for treatment; the significant difference observed among county incidence categories. While 88% (143) of dental staff received training in these step-by-step procedures, and 64% (103) in addition received a simulation, 24% (27) still reported deviations. The most popular disinfection product was 70% ethyl alcohol, used by 74% (125) of the respondents. The majority of dental staff did not use extraoral dental radiographs as an alternative to intraoral radiographs; a significant difference was observed among county incidence categories (see Table 3). 

All dental staff were asked four attitudinal statements regarding dental staff perception of risk and preparedness. The majority of respondents (1055, 85%) completely agreed/agreed that dental staff were at high risk of being infected by COVID-19. Sixty-two percent (766) perceived that their workplace had adequate infection control equipment, 84% (1035) experienced that their workplace handled the current situation well, while 41% (501) agreed that their workplace was well equipped to handle an escalation.

Table 4 shows the results of the multivariable regression analyses exploring associations between perception of risk and preparedness statements and selected independent variables. Less experienced dental staff, OR 2.0 (CI 1.4; 3.0), and dental staff in public practice, OR 2.4 (CI 1.3; 4.4), were more likely to perceive dental staff to have a high risk of being infected, while working in low incidence counties reduced odds, OR 0.5 (CI 0.3; 0.8), to perceive this risk. Dental staff in public sector, OR 0.3 (CI 0.2; 0.5) and those working at clinics not designated to treat patients suspected or confirmed to have COVID-19, OR 0.6 (0.4; 0.9) were less positive to preparedness of their workplace regarding infection control equipment. Dental staff in public sector, OR 0.2 (CI 0.1; 0.5), were less positive to how their workplace handled the current situation. 

Dental hygienists, OR 1.5 (CI 1.1; 2.2) and dental assistants, OR 1.4 (CI 1.0; 1.9), marginally, but statistically significantly associated with being positive to their workplace preparedness to handle an escalation, while dental staff at small clinics, OR 0.6 (CI 0.5; 0.9), public sector, OR 0.2 (CI 0.1; 0.4), and clinics not designated to treat patients suspected or confirmed to have COVID-19, OR 0.3 (CI 0.2; 0.4), were less positive to workplace preparedness.

## 4. Discussion

The COVID-19 pandemic is an unprecedented situation that has affected the population globally, especially healthcare workers, including dental staff. To the best of our knowledge, there are up to date 8 questionnaire studies that investigated COVID-19 outbreak and dentistry, summarized in Table 5.

None of the questionnaire studies assessed the urgent dental health care management and perception of risk and preparedness among the complete dental team, which includes not only dentists, but also dental hygienists and dental assistants. The appropriate infection prevention and control in order to limit the infection spread is a result of the efforts of the whole dental team.

In the present study, there was an over-representation of females and dental staff working in public sector, therefore the results should be interpreted with caution in this respect. In addition, the timing of a questionnaire is an important factor, because of the differences in pandemic peak in different countries and constantly changing guidelines. For example, in March 2020, CDC recommended that dental settings should prioritize urgent and emergency visits and delay elective visits. Already in April, some practices in the USA started reopening and providing the full range of dental health care. In Norway, from 16 March 2020 Health authorities recommended to reduce “one to one contact” in the dental setting by prioritize urgent care and delay elective care. Dental health service started gradual re-opening also for elective visits after national recommendations issued by the end of May 2020. The present questionnaire study was commenced 4/15 May 2020 and asked the information about the strictest confinement period in Norway, 13 March–17 April 2020. Therefore, the results of the present study may not be directly comparable with other studies, as for example the study by Kamate and co-workers was conducted much earlier [21]. Moreover, the respondents of the global surveys may have experienced different degrees of outbreak during the given survey time which possibly influenced their practices and perceptions.

In the present study, the majority of the respondents completely agreed/agreed that dental staff had a high risk of being infected by SARS-CoV-2. The New York Times magazine ranked dental staff among other healthcare workers to have the highest risk to be infected [29]. In Italy as well, the majority of respondents agreed that dentistry is a profession at risk [26,28]. On the contrary, only one out of five dentists perceived COVID-19 as very dangerous in Jordan [23]. It must be noted that the questionnaire among Jordanian dentists was distributed early in the global COVID-19 outbreak, when Jordan did not have any local cases, in addition to the fact that dentists in Jordan has experience with previous similar virus outbreaks. In the present study, the dental staff perception of a dental staff having a high risk of being infected positively associated with working in a public sector and having less professional experience, but negatively with working in low incidence counties.

The majority of the dental staff perceived that their clinic handled the current situation well, which negatively associated with working in public sector. However, less than a half of the respondents agreed that their workplace was well equipped to handle an escalation, which negatively associated with small clinics, clinics not designated to treat patients suspected or confirmed to have COVID-19 and also public sector. The differences in perceived preparedness between private and public sectors can be partly explained by differences in “locus of control”-while dentists working in private sector were solely themselves responsible for being prepared, while dental staff in public sector were part of a large organization andwere more dependent on decisions of others. As this was a questionnaire study, we do not know if they in fact were better prepared, but it seems they had a better confidence in perceiving their preparedness. There is reason to believe that the level of preparedness facing a virus outbreak like Sars-CoV-2 in a country or society is influenced by experience with earlier and similar epidemics, like MERS and SARS. Norway has not had a similar virus outbreak in the past and did not even have national recommendations for infection prevention and control in dental practice before 2018. Increased internationalization and prevalence of antibiotic resistance did then contribute to the development of recommendations, which were used as a foundation for organizing the activity in the dental health service during the COVID-19 outbreak.

To reduce the spread of Sars-CoV-2, The Norwegian Institute of Public Health recommended that all patients and accompanying persons should be clarified with regard to infection status and anamnesis per phone prior to their appointment [30]. The majority of the dental staff always/often inquired information per phone about symptoms and about history of travel, showing a high degree of compliance with the recommendations from the authorities. This finding is in line with the global questionnaire study [24] and a study from Italy, where phone triage, together with spaced appointments was the most commonly adopted precautionary measure, while deferring treatment in elderly and detecting body temperature in staff and patients were less commonly adopted precautionary measures [26]. The Jordanian study revealed limited comprehension of the extra precautionary measures, where a recommended procedure during the outbreak was to measure the temperature of staff and patients [23]. In the present study, the lower proportion of dental staff inquiring about symptoms and travel history were in high incidence counties. In addition, the lower proportion of dental staff in high incidence counties reported not to use prevention measures, such as mouth rinse before procedure, rubber dam and high-volume suction while treating a patient not suspected to have COVID-19. These results are in line with the Italian survey, where dentists from the highest prevalence areas reported to adopt preventive measures less frequently [26]. The authors suggested that the risk perception is lower in high incidence areas because it is more general. Therefore, risk perception in a dental clinic in high incidence areas is also lower. On the other hand, in the present study dental staff working in low incidence counties versus high incidence counties perceived dental staff as having a lower risk of being infected.

Teledentistry has been proposed only for conditions that could be managed by advice and managed or postponed by medication. It seems to be a useful platform to offer consultations when social distancing is warranted, to minimize direct patient interactions, and to reduce the use of personal protective equipment (PPE) as well as other highly valuable clinical resources during a pandemic [31]. A study evaluating the urgent dental care in North East of England in the first six weeks of the pandemic concluded that the phone triage system used to handle emergency and urgent dental care was both essential and effective [32]. In the present study, on average per week, five patients were treated in a dental office and twice as many received treatment by per phone, out of them one third received drug treatment for their dental condition. Thus, treatment per phone may be evaluated as effective also in Norway. In the present study, the most common conditions were severe dental pain from pulpal inflammation, abscess, or localized bacterial infection resulting in localized pain and swelling and pericoronitis or third-molar pain. This is in line with a study in Beijing, China, which reported that the utilization of emergency dental care decreased during COVID-19 outbreak and the distribution of the oral health conditions changed; more dental and oral infections were recorded, but less dental traumas compared to pre-COVID-19 period [33]. Moreover, the results of the present study are in line with a study from England, where the most frequent dental emergency conditions reported were acute pulpitis or periapical symptoms [32]. During the treatment of these conditions, the most aerosol generating procedures can be avoided.

In Norway, during the strictest confinement period, several public dental clinics were designated to provide urgent treatment for patients suspected or confirmed to have COVID-19. The number of private clinics that provided dental care to patients suspected or confirmed to have COVID-19 in Norway in this period is not currently known. Designated clinics were also implemented in the UK, where local urgent dental care hubs were arranged [34] and in China [35]. This was not the case in Italy where private sector provided much of the dental health service, and almost half of the private dentists reported to remain working during the outbreak [26].

In the present study, less than two thirds of the dental staff agreed that their workplace had adequate infection control equipment. Dental staff in public sector and those working at clinics not designated to treat patients suspected or confirmed to have COVID-19 were less positive to this statement. During the peak of the pandemic, the global stockpile of PPE was insufficient, and the demand for respirators and masks even for health care workers could not be met [36]. The majority of Italian dentists reported to have difficulties in finding needed PPE [27]. In the present study, up to 20% of the dental staff working in the clinics designated to treat patients suspected or confirmed to have COVID-19 reported not to have available PPE, such a respirators, gowns and aprons at their workplace. Even when treating patients not suspected to have COVID-19, 88% (389) of the dental staff working during the strictest confinement period in Norway applied additional infection control measures, though the WHO guidelines released 29 June for the health care advise that for patients not suspected to have COVID-19 standard precaution should be applied [37]. Every fifth responding dentist in Jordan reported that additional infection control measures, such as patients wearing masks and washing hands before getting into a dental chair, are not necessary and may create a panic [23]. The newly released (3 August 2020) WHO interim guidance for the provision of essential oral health services in the context of COVID-19 advises that all patients are encouraged to use medical or non-medical masks and practice hand hygiene on arrival and throughout the visit [38].

In the present study, almost all dental staff working in clinics that were not designated to treat patients suspected or confirmed to have COVID-19, knew where to refer a patient or where to find an information about it. The highest proportion of dental staff who did not know either clinics or where to find an information, were from high incidence counties. The majority of the respondents in the global survey were aware of the proper authority to contact in case a patient was suspected to have COVID-19 [24]. This demonstrates that dental staff were well informed, and thus potentially minimize the risk of infection spread.

In the present study, the majority of the dental staff working at the clinics designated to treat patients suspected or confirmed to have COVID-19, reported to follow local guidelines for additional infection prevention and control developed by county and university. According to the global survey, 90% of the respondents were updated with the current CDC or WHO guidelines for infection prevention and control [24]. Following guidelines is a crucial aspect in limiting infection spread. Dental treatment involves droplets and aerosol generating procedures, such as high-speed drills, dental hand-pieces, ultrasonic and air-flow devices, air-water syringe, ultrasonic scaler and oral prophylaxis cups/rotating brushes. A review has identified that SARS-CoV-2 may persist in the air in closed unventilated indoor areas for at least 30 min without losing infectivity [39]. Therefore, adequate time between patients in the dental office may minimize the risk of cross-infection. In the present study, 84% (144) of the dental staff working in clinics designated to treat patients suspected or confirmed to have COVID-19, reported to leave the room before the next patient for 35 min or more. Droplets and aerosols may contaminate surfaces, and it has been shown that viruses can sustain on surfaces for various time periods, depending on temperature and humidity, sometimes even up to 28 days [40]. Surface disinfection procedures with 62–71% ethanol, 0.5% hydrogen peroxide and 0.1% sodium hypochlorite seem to be the most effective against coronaviruses [40]. In the present study, the most common disinfection agent was reported to be 70% ethyl alcohol.

Mouth rinse before dental procedures has been shown to reduce microorganisms’ load in droplets and aerosols [41]. The most common mouth rinse is 0.02% chlorhexidine digluconate, which seems to be less effective against coronaviruses compared to hydrogen peroxide [40]. The majority of dental staff working in clinics designated to treat patients suspected or confirmed to have COVID-19 (93%, 50) reported to use mouth rinse (for example chlorhexidine digluconate or hydrogen peroxide) as an additional protective measure. High-volume suction was reported to be used by 85% (46) and rubber dam by 63% (34) of the dentists as an additional protective measure. According to the global survey, the majority of the respondents neither used mouth rinse nor rubber dam, but a proportion reported to have used high-volume suction [24]. Rubber-dam and high-volume suction are considered valid infection control measures during dental procedures and are recommended by American Dental Association in order to reduce aerosols during dental procedures [42,43,44,45,46]. Intraoral radiographic examination is the most common radiographic technique in dentistry, but as it may stimulate both saliva secretion and coughing, extraoral radiographs may be an appropriate alternative during a virus outbreak, but only a small proportion of dental staff working with patients suspected or confirmed to have COVID-19 reported to use extraoral radiographs [35].

The majority of respondents received training in the guidelines either digitally or in the clinics, which included training in putting on and removing PPE. Even though, 88% and 67% of the respondents reported to receive training and simulation, respectively, in step-by-step procedures for treatment, including PPE putting on and removing, 24% of the respondents working in clinics designated to treat patients suspected or confirmed to have COVID-19 reported the deviation in these procedures. This finding demonstrates that additional infection prevention and control procedures for treatment may not be easy to follow and require extra training. This calls for additional dental staff training in step-by-step procedures for dental treatment during an outbreak in order to minimize infection spread.

## 5. Conclusions

In general, urgent oral health care was managed relatively well in Norway and the majority of the dental staff perceived that their clinic handled the current situation well. However, only less than a half of the respondents agreed that their workplace was well equipped to handle an escalation. In the clinics designated to treat patients suspected or conformed to have COVID-19, lack of availability of several PPE was reported. Mainly local guidelines developed at a county level or universities were followed. Despite training and simulation in additional infection prevention and control step-by-step procedures, there were reported several deviations. Fewer dental staff in high incidence counties applied additional infection prevention measures compared to low and medium incidence counties.

The results of this study may be used to improve dental health service response to possible future outbreaks in Norway and other countries. The results call for additional staff training in using appropriate PPE and applying additional preventive measures for patients without and with infection.

## Figures and Tables

**Table 1 ijerph-17-05843-t001:** Background characteristics of study sample stratified by profession.

Background Characteristics	Dentists *N* = 590	Dental Hygienists *N* = 235	Dental Assistants *N* = 412	Total Respondents *N* = 1237
**Sex ***				
Female	468 (79%)	227 (97%)	411 (99.8%)	1106 (89%)
Male	122 (21%)	8 (3%)	1 (0.2%)	131 (11%)
**Age ***				
<30	73 (12%)	50 (21%)	35 (8%)	158 (13%)
30–40	229 (39%)	64 (27%)	75 (18%)	368 (30%)
41–50	151 (26%)	51 (22%)	110 (27%)	312 (25%)
51–60	87 (15%)	43 (18%)	124 (30%)	254 (21%)
>60	50 (8%)	27 (12%)	68 (17%)	145 (11%)
**Work experience ***				
0–9 years	216 (37%)	84 (36%)	89 (22%)	389 (31%)
>10 years	374 (63%)	151 (64%)	323 (78%)	848 (69%)
**Area of dental clinic ***				
Urban	274 (46%)	84 (36%)	138 (33%)	496 (40%)
Peri-urban	266 (45%)	132 (56%)	217 (53%)	615 (50%)
Rural	50 (9%)	19 (8%)	57 (14%)	126 (10%)
**Size of dental clinic**				
Large	436 (74%)	190 (81%)	318 (77%)	944 (76%)
Small	154 (26%)	45 (19%)	94 (23%)	293 (24%)
**In which sector is your main workplace ***				
Public	493 (84%)	234 (99.6%)	407 (99%)	1134 (92%)
Private	97 (16%)	1 (0.4%)	5 (1%)	103 (8%)
**County incidence categories ***				
High	103 (17%)	24 (10%)	38 (9%)	165 (13%)
Medium	311 (53%)	127 (54%)	216 (52%)	654 (53%)
Low	176 (30%)	84 (36%)	158 (38%)	418 (34%)
**Worked clinically with patients ***				
Yes	413 (70%)	66 (28%)	248 (60%)	727 (59%)
No	177 (30%)	169 (72%)	164 (40%)	510 (41%)

* *p* < 0.05 between different dental professions.

**Table 2 ijerph-17-05843-t002:** Organization of urgent dental care in dental health service and management of patients NOT suspected to have COVID-19 during 13 March 2020–17 April 2020, stratified by county incidence categories.

Urgent Dental Care Questions	Low Incidence Counties	Medium Incidence Counties	High Incidence Counties	Total Number of Responses
**Information about patients inquired per phone ***				
Asking about symptoms (fever, cough, shortness of breath)				
Always	378 (90.4%)	554 (84.7%)	126 (76.4%)	1058 (85.5%)
Often	30 (7.2%)	71 (10.9%)	23 (13.9%)	124 (10.0%)
Seldom	1 (0.2%)	21 (3.2%)	10 (6.1%)	32 (2.6%)
Never	5 (1.2%)	4 (0.6%)	2 (1.2%)	11 (0.9%)
I do not know	4 (1.0%)	4 (0.6%)	4 (2.4%)	12 (1.0%)
Asking about history of travel to affected areas				
Always	337 (80.6%)	488 (74.6%)	115 (69.7%)	940 (76.0%)
Often	52 (12.4%)	86 (13.1%)	26 (15.8%)	164 (13.3%)
Seldom	14 (3.3%)	56 (8.6%)	8 (4.8%)	78 (6.3%)
Never	8 (1.9%)	12 (1.8%)	4 (2.4%)	24 (1.9%)
I do not know	7 (1.7%)	12 (1.8%)	12 (7.3%)	31 (2.5%)
**How many patients NOT suspected to have COVID-19 have you treated (average per week)**				
Mean (SD)	4.24 (4.7)	4.52 (3.5)	4.94 (7.2)	4.50 (4.6)
Median (range)	3.00	4.00	3.00	3.00
**How many patients have you treated per phone (average per week)**				
Mean (SD)	12.32	9.42	11.11	10.58
Median (range)	10.00	6.00	5.00	7.00
**How many patients on average treated per phone received drug treatment (average per week)**				
Mean (SD)	2.85	2.63	2.26	2.63
Median (range)	2.00	2.00	1.00	1.00
**When treating patients NOT suspected to have COVID-19, did you apply additional infection control measures**				
Yes	117 (86.0%)	217 (90.4%)	55 83.3%)	389 (88.0%)
No	12 (8.8%)	20 (8.3%)	9 (13.6%)	41 (9.3%)
I do not know	7 (5.1%)	3 (1.3%)	2 (3.0%)	12 (2.7%)
**How do you clean and disinfect room and equipment after having a patient ***				
70% ethyl alcohol	394 (94.3%)	592 (90.5%)	135 (81.8)	1121 (90.6%)
Sodium hypochlorite at 0.5% (5000 ppm)	6 (1.4%)	3 (0.5%)	2 (1.2%)	11 (0.9%)
Sodium hypochlorite at 0.1% (1000 ppm)	8 (1.9%)	9 (1.4%)	5 (3.0%)	22 (1.8%)
Potassium peroxymonosulfate (Virkon)	21 (5.0%)	16 (2.4%)	24 (14.5%)	61 (4.9%)
Disodium carbon (hydrogen peroxide (Perasafe))	32 (7.7%)	19 (2.9%)	25 (15.2%)	76 (6.1%)
Other	396 (94.7%)	591 (90.4%)	145 (87.9%)	1132 (91.5%)
**Which personal protection equipment did you use when treating patients**				
Respirators FFP2 or FFP3 standard or equivalent	27 (19.9%)	53 (22.1%)	21 (31.8%)	101 (22.9%)
Gowns	79 (58.1%)	135 (56.3%)	35 (53.0%)	249 (56.3%)
Surgical caps	107 (78.7%)	195 (81.3%)	50 (75.8%)	352 (79.6%)
Medical masks	132 (97.1%)	238 (99.2%)	63 (95.5%)	433 (98.0%)
Gloves	132 (97.1%)	239 (99.6%)	64 (97.0%)	435 (98.4%)
Eye protection (googles or face shield)	132 (97.1%)	233 (97.1%)	64 (97.0%)	429 (97.1%)
Apron	73 (53.7%)	136 (56.7%)	35 (53.0%)	244 (55.2%)
**Did you use any of the following as an additional protective measure while treating patients**				
Mouth rinse before a dental procedure *	106 (78.5%)	208 (87.0%)	50 (75.8%)	364 (82.7%)
Rubber-dam *	76 (56.3%)	99 (41.4%)	22 (33.3%)	197 (44.8%)
High volume suction *	111 (82.2%)	209 (87.4%)	49 (74.2%)	369 (83.9%)
Chemo-mechanical caries removal	5 (3.7%)	9 (3.8%)	4 (6.1%)	18 (4.1%)
Scheduling a patient where high-speed handpiece needs to be used as the last patient of the day	52 (38.5%)	109 (45.6%)	23 (34.8%)	184 (41.8%)
Absorbable suture	44 (32.6%)	76 (31.8%)	23 (34.8%)	143 (32.5%)
Extraoral dental radiographs as an alternative to intraoral radiographs	19 (14.1%)	28 (11.7%)	8 (12.1%)	55 (12.5%)
**Is your clinic designated to treat patients suspected or confirmed to have COVID-19? ***				
Yes	94 (22.5%)	54 (8.3%)	22 (13.3%)	170 (13.7%)
No	324 (77.5%)	600 (91.7%)	143 (86.7%)	1067 (86.3%)
**If not, what do you do if such a patient requires urgent treatment ***				
I know clinics that are eligible to treat such patients	319 (98.5%)	593 (98.5%)	129 (90.2%)	1041 (97.4%)
I do not know where such clinics are, but I know where to find the information on how to find them	5 (1.5%)	8 (1.3%)	10 (7.0%)	23 (2.2%)
I do not know where such clinics are and where to find the information on how to find them	0 (0.0%)	1 (0.2%)	4 (2.8%)	5 (0.5%)

* *p* < 0.05 between different counties of incidence.

**Table 3 ijerph-17-05843-t003:** Organization of urgent dental care for patients suspected or confirmed to have COVID-19 performed during 13 March, 2020–17 April, 2020, stratified by county incidence categories.

Urgent Dental Care for Patients Suspected or Confirmed to Have COVID-19	Low Incidence Counties	Medium Incidence Counties	High Incidence Counties	Total Number of Responses
**How many patients suspected or confirmed to have COVID-19 have you treated (average per week)**				
Mean (SD)	0.05 (0.3)	0.03 (0.2)	0.8 (4.1)	0.05 (0.3)
Median (range)	0 (0; 3)	0 (0; 2)	0 (0; 32)	0 (0; 32)
**How many patients suspected or confirmed to have COVID-19 have you treated (average per week)**				
None	91 (96.3%)	53 (97.9%)	19 (89.4%)	161 (94.5%)
1–3	3 (3.7%)	4 (2.1%)	2 (7.5%)	7 (4.4%)
More than 3	0 (0%)	0 (0%)	1 (3.1%) *	2 (1.0%)
**How long do you leave the room in between patients ***				
No limit	2 (2.1%)	0 (0.0%)	0 (0.0%)	2 (1.2%)
35 min–3 h	47 (50.0%)	10 (18.2%)	10 (45.5%)	67 (39.2%)
>3 h	11 (11.7%)	16 (29.1%)	2 (9.1%)	29 (17.0)
Until the next day	21 (22.3%)	19 (34.5%)	6 (27.3%)	46 (26.9%)
I do not know	13 (13.8%)	10 (18.2%)	4 (18.2%)	27 (15.8%)
**Which guidelines did you follow**				
Developed at county level	52 (55.3%)	23 (42.6%)	9 (40.9%)	84 (49.4%)
Developed by a university	25 (26.6%)	22 (40.7%)	12 (54.5%)	59 (34.7%)
I do not know	7 (7.4%)	2 (3.7%)	0 (0.0%)	9 (5.3%)
Other	10 (10.6%)	7 (13.0%)	1 (4.5%)	18 (10.6%)
**Have you received a training in additional infection prevention and control guidelines ***				
Yes, digitally	128 (69.2%)	91 (55.8%)	21 (47.7%)	240 (61.2%)
Yes, at the clinic	41 (22.2%)	30 (18.4%)	17 (38.6%)	88 (22.4%)
No	16 (8.6%)	42 (25.8%)	6 (13.6%)	64 (16.3%)
**Has your clinic developed step-by-step procedures for treatment ***				
Yes	93 (98.9%)	47 (87.0%)	22 (100.0%)	162 (95.3%)
No	0 (0.0%)	6 (11.1%)	0 (0.0%)	6 (3.5%)
I do not know	1 (1.1%)	1 (1.9%)	0 (0.0%)	2 (1.2%)
**Have you received a training in step-by-step procedures for treatment**				
Yes	86 (92.5%)	37 (78.7%)	20 (90.9%)	143 (88.3%)
No	7 (7.5%)	10 (21.3%)	2 (9.1%)	19 (11.7%)
**Have you received a simulation in step-by-step procedures for treatment**				
Yes	60 (64.5%)	31 (66.0%)	12 (54.5%)	103 (63.6%)
No	33 (35.5%)	16 (34.0%)	10 (45.5%)	59 (36.4%)
**Have you experienced a deviation from step-by-step procedures**				
Yes	12 (17.9%)	12 (35.3%)	3 (23.1%)	27 (23.7%)
No	55 (82.1%)	22 (64.7%)	10 (76.9%)	87 (76.3%)
**Have you been in need of an updated scientific information regarding treatment**				
Yes	234 (71.6%)	364 (66.8%)	102 (71.3%)	700 (69.0%)
No	93 (28.4%)	181 (33.2%)	41 (28.7%)	315 (31.0%)
**If yes, was the information easily accessible**				
Yes	57 (83.8%)	31 (77.5%)	12 (92.3%)	100 (82.6%)
No	11 (16.2%)	9 (22.5%)	1 (7.7%)	21 (17.4%)
**Which personal protection equipment was available at your workplace**				
Respirators FFP2 or FFP3 standard or equivalent *	87 (92.6%)	42 (77.8%)	19 (86.4%)	148 (87.1%)
Gowns	90 (95.7%)	45 (83.3%)	21 (95.5%)	156 (91.8%)
Surgical caps	92 (97.9%)	49 (90.7%)	21 (95.5%)	162 (95.3%)
Medical masks	94 (100%)	52 (96.3%)	21 (95.5%)	167 (98.2%)
Gloves	94 (100.0%)	52 (96.3%)	21 (95.5%)	167 (98.2%)
Eye protection (googles or face shield)	93 (98.9%)	52 (96.3%)	21 (95.5%)	166 (97.6%)
Apron *	83 (11.7%)	35 (64.8%)	18 (81.8%)	136 (80.0%)
**Which personal protection equipment did you use when treating patients**				
Respirators FFP2 or FFP3 standard or equivalent	61 (89.7%)	35 (92.1%)	13 (100.0%)	109 (91.6%)
Gowns	65 (95.6%)	36 (94.7%)	13 (100.0%)	114 (95.8%)
Surgical caps	65 (95.6%)	36 (94.7%)	13 (100%)	114 (95.8%)
Medical masks	56 (82.4%)	34 (89.5%)	12 (92.3%)	102 (85.7%)
Gloves	66 (97.1%)	7 (97.4%)	13 (100.0%)	116 (97.5%)
Eye protection (googles or face shield)	66 (97.1%)	36 (94.7%)	13 (100.0%)	115 (96.6%)
Apron	28 (41.2%)	20 (52.6%)	6 (46.2%)	54 (45.4%)
**Did you use any of the following as an additional protective measure while treating patients**				
Mouth rinse before procedure	23 (85.2%)	19 (100.0%)	8 (100.0%)	50 (92.6%)
Rubber-dam	19 (70.4%)	12 (63.2%)	3 (37.5%)	34 (63.0%)
High-volume saliva ejectors	4 (14.8%)	2 (10.5%)	2 (25.0%)	8 (14.8%)
Chemo-mechanical caries removal	1 (3.7%)	1 (5.3%)	1 (12.5%)	3 (5.6%)
Scheduling a patient where high-speed handpiece needs to be used as the last patient of the day	20 (74.1%)	14 (73.7%)	6 (75.0%)	40 (74.1%)
Absorbable suture	16 (59.3%)	7 (36.8%)	5 (62.5%)	28 (51.9%)
Extraoral dental radiographs as an alternative to intraoral radiographs *	8 (29.6%)	2 (10.5%)	5 (62.5%)	15 (27.8%)
**How do you clean and disinfect room and equipment after having a patient**				
70% ethyl alcohol	67 (71.3%)	41 (75.9%)	17 (77.3%)	125 (73.5%)
Sodium hypochlorite at 0.5% (5000 ppm)	5 (5.3%)	2 (3.7%)	0 (0%)	7 (4.1%)
Sodium hypochlorite at 0.1% (1000 ppm)	3 (3.2%)	2 (3.7%)	0 (0%)	5 (2.9%)
Potassium peroxymonosulfate (Virkon)	34 (36.2%)	20 (37.0%)	4 (18.2%)	58 (34.1%)
Disodium carbon with hydrogen peroxide (Perasafe)	24 (25.5%)	13 (24.1%)	8 (36.4%)	45 (26.5%)
Do not know	11 (11.7%)	6 (11.1%)	3 (13.6%)	20 (11.8%)
Other	9 (9.6%)	9 (16.7%)	3 (13.6%)	21 (12.4%)

* *p* < 0.05.

**Table 4 ijerph-17-05843-t004:** Multivariable regression analyses on associations between dental staff perception of risk and preparedness, and selected independent variables.

Independent Variables	Dental Staff Have a High Risk of Being Infected (Completely Agree/Agree) Adjusted ^i^ OR (95%CI)	If Workplace Has Adequate Infection Control Equipment (Completely Agree/Agree)Adjusted ^ii^ OR (95%CI)	If Workplace Handles Current Situation (Completely Agree/Agree)Adjusted ^iii^ OR (95%CI)	Workplace is Well Equipped to Handle an Escalation (Completely Agree/Agree) Adjusted ^iv^ OR (95%CI)
**Sex**				
Male (ref)	1	1	1	1
Female	0.9 (0.5; 1.5)	1.3 (0.8; 1.9)	0.9 (0.5; 1.6)	0.8 (0.5; 1.2)
**Work experience**				
>10 years (ref)	**1**	1	1	1
0–9 years	**2.0 (1.4; 3.0)**	0.9 (0.7; 1.1)	0.9 (0.6; 1.2)	1.0 (0.8; 1.4)
**Profession**				
Specialist/General dental practitioner (ref)	1	1	1	**1**
Dental hygienist	0.8 (0.5; 1.3)	1.3 (0.9; 1.8)	1.6 (1.0; 2.5)	**1.5 (1.1; 2.2)**
Dental assistant	1.0 (0.7; 1.5)	1.1 (0.8; 1.5)	1.5 (1.0; 2.1)	**1.4 (1.0; 1.9)**
**Area of dental clinic**				
Urban (ref)	1	1	1	1
Peri-urban	1.3 (0.9; 1.8)	0.9 (0.7; 1.1)	0.9 (0.6; 1.2)	1.1 (0.8; 1.4)
Rural	1.5 (0.7; 2.9)	0.9 (0.6; 1.4)	1.0 (0.6; 1.9)	1.2 (0.7;1.9)
**Size of dental clinic**				
Large (ref)	1	1	1	1
Small	1.1 (0.7; 1.7)	0.9 (0.7; 1.2)	0.9 (0.6; 1.4)	**0.6 (0.5; 0.9)**
**In which sector is your main workplace**				
Private (ref)	**1**	**1**	**1**	**1**
Public	**2.4 (1.3; 4.4)**	**0.3 (0.2; 0.5)**	**0.2 (0.1; 0.5)**	**0.2 (0.1; 0.4)**
**County incidence categories**				
High (ref)	1	1	1	1
Medium	0.6 (0.3; 1.2)	1.3 (0.9; 2.0)	1.4 (0.8; 2.3)	1.4 (0.9; 2.1)
Low	**0.5 (0.3; 0.8)**	1.3 (0.9; 1.8)	1.4 (0.8; 2.2)	1.3 (0.8; 1.9)
**Worked clinically with patients during COVID-19 outbreak 13 March–17 April 2020**				
Yes (ref)	1	1	1	1
No	0.8 (0.5; 1.1)	0.9 (0.7; 1.1)	1.2 (0.9; 1.7)	1.1 (0.8; 1.4)
**Is your clinic designated to treat patients suspected or confirmed to have COVID-19?**				
Yes (ref)	1	**1**	1	**1**
No	1.3 (0.8; 2.1)	**0.6 (0.4; 0.9)**	0.9 (0.5; 1.4)	**0.3 (0.2; 0.4)**

(ref)—reference category, significant associations (*p* < 0.05) presented in bold. ^i^ Adjusted for variables that resulted in statistically significant associations according to univariable analyses. ^ii^ Adjusted for variables that resulted in statistically significant associations according to univariable analyses. ^iii^ Adjusted for variables that resulted in statistically significant associations according to univariable analyses. ^iv^ Adjusted for variables that resulted in statistically significant associations according to univariable analyses.

**Table 5 ijerph-17-05843-t005:** The summary of the published questionnaire studies investigating COVID-19 outbreak and dentistry. The studies are presented in chronological order of study date; data searched was finished 4 July 2020.

First Author	Country	Date	Participants	Sample Size	Aim
Kamate [21]	Global Mostly represented by participants from Asia and Americas	December 25, 2019–February 20, 2020	Dentists	860	Knowledge, attitudes and practices
Quadri [22]	Saudi Arabia	March 2020	Dental interns, dental auxiliaries, dental specialists	706	Current knowledge and plausible misconceptions
Khader [23]	Jordan	March 2020	Dentists	368	Awareness, perception and attitude
Ahmed [24]	Global Mainly represented by Pakistan, Saudi Arabia, China and India	March 10–17, 2020	Dentists	650	Fear and practice modification
Duruk [25]	Turkey	March 16–20, 2020	Dentists	1958	Clinical attitudes and behaviors
Cagetti [26]	Italy	April 2020	Dentists	3599	Symptoms/signs, protective measures, awareness, and perception
Consolo [27]	Italy	April 2–21, 2020	Dentists	356	Behaviors, emotions and concerns
De Stefani [28]	Italy	April 11–18, 2020	Dentists	1500	Knowledge, perception and attitude in treating potentially infected patients

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
