# Peer review of "Dental Health Services Response to COVID-19 in Norway"

_ijerph, 2020, doi:10.3390/ijerph17165843_

Round 1

Reviewer 1 Report

Dear authors, kindly find some comments of your article.

  1. There is a couple of grammar mistakes and typos.
  2. The study design is well prepared but with lot of variables which make the study confusing and unclear.
  3. I would recommend to use less variables and better compare the correlation between public sector and private sector. The attitude to the preventive measures and risk perception between these two might be different.
  4. It is not clear, if dental institutions designated to treat patients suspected or confirmed to have COVID-19 were public or private and if they provide emergency treatment only or standard dental treatment as well.
  5. In case of urgent care need I cannot imagine any teledentistry. How this was conducted (acute periodontal infection or cellulitis treated by phone?)
  6. In the results section the data should be unified (numerical explanation, %). 
  7. In the discussion section there are several paragraphs repeating (r.315 - 340)
  8. In some dental conditions intra-oral radiographic examination are crucial for proper diagnosis. I do not consider extra-oral radiograps as  appropriate alternative.
  9. The sentence 395 - 397 is confusing.

Reviewer 2 Report

The article Response to COVID-19 in Dental Health Service in Norway deals with up to date topic, but i am not sure about the long term usefulness. The data collection and statistical analysis sounds strong and well evaluated.  For me as dentist working in Czech republic the article is not so interesting, but on the other hand I believe that for many other professions (e.g. epidemiologist, crisis management etc.) can be very usefull. Surely there is good potential for citation moreover if other countries will provide similar studies. There are some minor spelling mistakes and some parts seems to be repeated twice within the article. the article contains exhaustingly much results. I do not recommend deletion of any results, cause all of them can be usefull for some readers, but the authors can provide slightly more deduction from the results and lead the reader.

Reviewer 3 Report

I recommend this manuscript to be accepted for publication upon minor revisions.

This is a contemporary and important piece of research that can inform the way dental practices respond to future pandemics in Norway and globally. The research makes a significant contribution to the literature on dental practices during outbreaks.

The manuscript is generally well-presented. The authors provide a clear background into the context of the research, highlighting the relevance of the pandemic to dental practices within the Norwegian context. There is a well-defined gap in the literature for the need of exploring how urgent dental practices were managed during an intense period of the pandemic in Norway. The aim of the paper is clearly stated. 

There is sufficient detail provided to describe the study design and participants. The authors also provide a thorough description of the questionnaire used in the cross-sectional study. The statistical analyses techniques used are described well; however, some justification or reference is needed for the cut-off of p<0.2 used in the bivariate analyses to include variables in the final multiple regression model. It would also be helpful to provide the basis for selecting variables that were considered as ‘potential determinants’ of the outcomes investigated.  

The results are well-summarised in tables and explained in texts. There are, however, few explanations/ clarifications needed to improve the presentation and understanding of some of the results.

The significance testing for the demographic variables in Table 1 need some clarification – were the differences tested between demographic variables using the total number of respondents or between the different dental professions?  

There are too many references to Table 2 within lines 140-157. This is also the case for references to Table 3 from lines 166 to 187. A statement can be made at the beginning of the paragraphs where these results are presented to show that their sources are from these Tables.  

It should be clarified that the odds ratio (OR) for the likelihood of dental hygienist and dental assistants being more positive to the workplace preparedness to handle an escalation could have been due to chance (looking at the confidence intervals).

Table 4: OR for association between dental assistant (type of profession) and workplace having adequate infection control equipment should be written as ‘1.1’ and not ‘1,1’.

The results of the research are well-discussed and related to previous similar research findings. The conclusions are sound and viable recommendations are also made for policy and practice. There is no need for Table 5 in the discussion; as references can be made to these studies without having them in a table. There are some repetitions in the discussion on the Jordan study and the WHO’s guidelines. See lines 316-320 and lines 334-338.

There are several minor grammatical and spelling errors that need to be corrected. Some of these are highlighted below:

  • Line 18 – change ‘conformed’ to ‘confirmed
  • Line 21 – change “staff had a high risk for being infected” to “staff had a high risk of being infected”
  • Line 113 – change ‘groups’ to ‘grouped
  • Line 118 – Anova is an abbreviation; so, change to capitals (ANOVA) and define in full
  • Line 127 – change ‘voluntarily’ to ‘voluntary’
  • Line 135 – change ‘positive tested’ to ‘tested positive’
  • Lines 140 – 142: sentence is a bit confusing – break down into two separate sentences
  • Lines 143-145: I can understand the point being made; but sentence should be revised for more clarity.
  • Line 171: ‘...4% in low incidence counties.’
  • Line 199: ‘...to have a risk to be infected,...’ change to ‘...of having a high risk of being infected,...’
  • Line 200: change ‘perceive’ to ‘perceived’
  • Lines 241-244: Very long sentence – authors should consider revising
  • Line 269: change ‘companying’ to ‘accompanying’.
  • Line 279: change ‘where’ to ‘were’.
  • Line 284-285: sentence not very clear and needs rephrasing.
  • Line286: ‘...low incidence counties versus high incidence counties perceived dental staff as having a lower risk of being infected’.
  • Line 291: ‘...North East of England in the first six weeks...’.
  • Line 293: ‘In the present study, on average per week, 5...’
  • Lines 298-303: a very long sentence - break sentence into two/ three separate sentences
  • Line 328: ‘...reported not to have available some ..’ – delete the word ‘some’.
  • Line 343: ‘...to contact in case of a patient was suspected to ...’ – delete ‘of’
  • Line 344: ‘This demonstrates that dental staff were well informed,...’
  • Line 358: ‘...may persist in the air in closed unventilated ...’
  • Lue 359-360: ‘...may minimize the risk of cross-infection’
  • Line 361: change ‘conformed’ to ‘confirmed’
  • Line 375: ‘...reported to have used high volume suction...’
  • Line 382: ‘...majority of respondents received a..’ – delete ‘a’
  • Line 383: ‘It is also advice staff training to put on and removing PPE’. – revise sentence
  • Line 392: ‘The majority of respondents agreed that dental staff have a high risk of being infected’.
  • Line 399: ‘There was reported a lack of availability...’ – delete ‘a’
